# Comorbidities in osteoarthritis (ComOA): a combined cross-sectional, case–control and cohort study using large electronic health records in four European countries

Subhashisa Swain [ORCID] ,[1,2] Anne Kamps,[3] Jos Runhaar,[3] Andrea Dell'Isola [ORCID] ,[4] Aleksandra Turkiewicz [ORCID] ,[4] Danielle Robinson [ORCID] ,[5] V Strauss,[5] Christian Mallen,[6] Chang-Fu Kuo [ORCID] ,[7] Carol Coupland,[8] Michael Doherty,[1,9] Aliya Sarmanova,[10] Daniel Prieto-Alhambra [ORCID] ,[5] Martin Englund,[4] Sita M A Bierma-Zeinstra,[11] Weiya Zhang[1,9]

For numbered affiliations see end of article.

**Correspondence to**
Professor Weiya Zhang;
weiya.zhang@nottingham.ac.uk

## ABSTRACT

**Introduction** Osteoarthritis (OA) is one of the leading chronic conditions in the older population. People with OA are more likely to have one or more other chronic conditions than those without. However, the temporal associations, clusters of the comorbidities, role of analgesics and the causality and variation between populations are yet to be investigated. This paper describes the protocol of a multinational study in four European countries (UK, Netherlands, Sweden and Spain) exploring comorbidities in people with OA.

**Methods and analysis** This multinational study will investigate (1) the temporal associations of 61 identified comorbidities with OA, (2) the clusters and trajectories of comorbidities in people with OA, (3) the role of analgesics on incidence of comorbidities in people with OA, (4) the potential biomarkers and causality between OA and the comorbidities, and (5) variations between countries.

A combined case–control and cohort study will be conducted to find the temporal association of OA with the comorbidities using the national or regional health databases. Latent class analysis will be performed to identify the clusters at baseline and joint latent class analysis will be used to examine trajectories during the follow-up. A cohort study will be undertaken to evaluate the role of non-steroidal anti-inflammatory drugs (NSAIDs), opioids and paracetamol on the incidence of comorbidities. Mendelian randomisation will be performed to investigate the potential biomarkers for causality between OA and the comorbidities using the UK Biobank and the Rotterdam Study databases. Finally, a meta-analyses will be used to examine the variations and pool the results from different countries.

**Ethics and dissemination** Research ethics was obtained according to each database requirement. Results will be disseminated through the FOREUM website, scientific meetings, publications and in partnership with patient organisations.

## Strengths and limitations of this study

► This is first ever multicentre study on comorbidities in osteoarthritis in Europe involving nearly 27 million electronic health records.
► More than 60 chronic conditions are being studied—representing a wider coverage of diseases.
► We will examine the causal association using Mendelian Randomisation -Phenome-Wide Association Study methods with genetic data collected from two countries
► Same protocol with robust statistical methods will be used across all countries to replicate the findings and examine the variations.
► Possible biases may be introduced by the nature of electronic health records and length of data availability.

## BACKGROUND

Osteoarthritis (OA) affects 27% of people aged over 45 years at peripheral synovial joints such as knees, hips, hands and feet.[1] It is by far the most common form of arthritis, and a leading cause of chronic joint pain and disability in older people.[2 3] It is anticipated that the burden of OA will continue to rise in the coming decades because of population ageing and the increasing obesity prevalence—two major risk factors for OA.[4 5] Co-occurrence of multiple chronic conditions in an individual with ageing is becoming a norm and OA is not an exception to this.

A recent systematic review has confirmed that people with OA are more likely to have other diseases, especially stroke, peptic ulcer, hypertension and depression.[6] Vast majority of these studies focused on additional

presence (comorbidity) of cardiovascular and musculoskeletal (MSK) conditions only.[7–9] Whether these comorbidities just co-exist with OA, share common risk factors with OA or are causes or consequences of OA remains largely unknown. There was also reporting of wide heterogeneity in definitions of OA and other chronic conditions, diagnosis and recording of diseases, sample sizes and number of diseases studied in previously published studies included in the review.[6] This diversity made the comparison and pooled estimation of comorbidity prevalence difficult.

Comorbidity in OA can occur due to multiple factors. Various hypotheses have been used to explain the existence of comorbidities in general, the most accepted of which are the concordant (diseases sharing similar pathophysiological risk factors) and discordant (diseases not sharing similar pathophysiological risk factors) theories.[10] Additionally, prescription of drugs is also reported to be associated with comorbidity and multimorbidity.[11] Especially in people with OA, the prescription of analgesics is common, and is associated with increased risk of other conditions such as cardiovascular, gastrointestinal (GI) and chronic renal diseases.[12 13] Also, having multiple chronic conditions increases the chances of polypharmacy which further escalates the risk of other conditions.

OA is one of the leading conditions reported in multimorbidity research. Exploring the association of OA with other diseases would help in further explaining the burden and pattern of the comorbidity.[14] However, the major issue in OA comorbidity research is the low number and specific types of conditions studied.[15] Therefore, it is important to develop a consensus on both the count and typology of conditions to be studied to enable comparisons across populations and to derive pooled estimates as appropriate. Further, using uniform methods and definitions of diseases in computing these estimates would reduce heterogeneity and make the comparison more reliable.

Understanding the temporal association with comorbidity and disease trajectory is crucial for any chronic condition, and this is possible through studies using longitudinal databases.[16] However, one of the limitations of using observational data is that causal associations are difficult to establish, due to the interference of known and unknown confounders. In this study, we have used the more recently developed method 'Mendelian Randomisation' (MR) that can determine causal estimates through combining the use of genetic data and instrumental variable (IV) methods.

The burden of diseases in primary care often depends on the population structure, health infrastructure and individual factors such as income and education. Such factors vary between countries, because of the heterogeneity mentioned above there are no robust data comparing OA and its comorbidities between countries. Therefore, the aims of this study were to explore the burden, pattern and causal factors of comorbidities in people with OA across four European countries using national registration databases in the UK, the Netherlands, Sweden and Spain.

## OBJECTIVES
1. To estimate the prevalence, incidence, and time sequence of comorbidities in OA.
2. To examine the clusters of comorbidities and trajectories of clusters in OA and associations with death.
3. To investigate the associations between commonly used OA drugs, such as NSAIDs and opioids, and risk of comorbidities.
4. To identify the potential biomarkers and causal pathways between OA and the comorbidities.
5. To examine the variations of OA comorbidities and clusters across countries.

## METHODS
### Databases
Four routinely collected national (the UK and the Netherlands) or regional (Sweden and Spain) health databases will be used for objectives 1–3. In addition, for objective 4, genomic associations of OA with comorbidities will be examined using two cohort studies from the UK (UK Biobank)[17] and the Netherlands (Rotterdam study).[18] The four national representative and regional databases contain information about the population with primary care consultations in four different countries. The longitudinal databases provide information about the diagnosis of the diseases by the general practitioners (GPs) and some diagnoses made in secondary care, prescription of drugs, deaths and other health utilisation indicators. The details of the databases and their properties are given in tables 1 and 2.

### Participants
People registered with the respective databases aged 18 years or above are eligible for the study.

### Patient public involvement
Three patient and public involvement representatives (with OA and multiple chronic conditions) were involved in this study through group meetings. Difficulties of living with multiple conditions, lack of research in causal relationship and identification of diseases to be studied and the role of drugs in comorbidity were discussed. They are constantly in touch through providing their inputs at each step of the study.

### Definition of OA
OA will be defined as having at least one recorded physician diagnosis of OA for hip, knee, ankle/foot, wrist/hand or site recorded as 'unspecified' during the study period for the respective database. People with any previous recording of the OA prior to the start date of the study will be excluded.

### Comorbidities
We defined comorbidity as the recording of diagnosis of predefined chronic conditions in individuals using either

**Table 1** Characteristics of the included databases

| | Netherlands | Spain | Sweden | UK |
|---|---|---|---|---|
| **Objectives 1–3** | | | | |
| Name of the database | Integrated Primary Care Information | The Spanish Sistema information del Deveolpment de l'Investigació a Atenció Primària | Skåne Healthcare Register | Clinical Practice Research Datalink |
| Settings | Routinely collected primary care database | Routinely collected primary care data | Swedish healthcare in Skane region, primary, specialist and in-patient care | Routinely collected primary care database with linkage database |
| Size and coverage | 2.2 million (Randomly distributed over the country) | 6.5 million (>85% of total Catalan region) | 1.3 million (all residents of the Skane region) | 17 million (country-wide, nearly 740 practices) |
| Start year | 1998- (Better coding after 2000) | 2006 | 1998 | 1993- |
| Age group | All | All | All | All |
| Gender | All | All | All | All |
| Coding system | ICPC | ICD 10 | ICD 10 | Read codes and ICD 10 |
| Drug prescribed by | GP | GP | GP | GP |
| Death record (Either date of death and/or cause) | Both date and cause | Only date | Both date and cause (until year 2015) | Only date |
| Covariates/ additional variables | NA | BMI, smoking, alcohol, social class, cholesterol, and other biomarkers | Education, Income, profession, and sick leave, residential area, region of birth | BMI, smoking, alcohol, deprivation index, ethnic group |

BMI, Body mass index; GP, general practitioners; ICD, International Classification of Diseases, Tenth Edition; ICPC, International classification of primary care; NA, Not available.

International Classification of Diseases, 10th Edition (ICD-10) or Read or international classification of primary care (ICPC) code. An extensive list of 61 chronic conditions was prepared from the Quality Outcome Framework,[19] list of the US Department of Health and Human Services Initiative on Multiple Chronic Conditions,[20] global burden of diseases[21] and the Charlson Comorbidity Index.[22] The list has been updated with findings from our systematic review[6] and a previous UK community-based knee pain study[6 23] by including common and important morbidities not included in the above.[24 25] A code mapping exercise was conducted to finalise the list of conditions available for all the research centres. The detailed list of the conditions is given in table 3.

## Covariates

Age and sex will be used in all centres as covariates to adjust in regression models. Additionally, information on body mass index (BMI), smoking, alcohol use, socioeconomic variables such as education level, income, place of birth (to identify those who immigrated to the country), and residential area, marriage (or registered partner) will be included when available. For calculating severity of the comorbidities in an individual, Elixhauser comorbidity index will be used to estimate the impact of comorbidities on death.[26 27] Missing data on covariates will be substituted using multiple imputation methods, provided that the data is missing at random, if applicable.

**Table 2** Database for the mendelian randomisation

| Name of the database | Rotterdam cohort study | UK biobank |
|---|---|---|
| Population coverage | 15 000 | 500 000 |
| Age group | ≥40 years | 40–69 years |
| Start year -till now | 1989–onwards | 2010- |
| Types of data | Radiographic data, joint pain, joint stiffness, of hip, knee and hand, GWAS, biochemical markers | Genetic and phenotypes |

GWAS, Genome Wide Association Studies.

**Table 3**  List of chronic conditions across four databases

| Sl no | Conditions | ICD 10 | ICPC | Read code |
|---|---|---|---|---|
| 1 | Anaemia (All types) | D50-D64 | B78, B80, B81, B82 | D00, D01… |
| 2 | Ankylosing spondylitis | M45.9 | NA | N10… |
| 3 | Anxiety disorder | F41.0-F41.9 | P74, P74.01, P74.02 | E200…, Eu41… |
| 4 | Asthma | J450-J45.9, J46.9 | R96… | 66Y…, H33… |
| 5 | Benign prostatic hypertrophy | N40.9 | Y85 | K20…, K21…, K22… |
| 6 | Cardiac arrhythmias (Atrial Fibrillation) | I47.0-I49.9 | K78 | Gyu…, G573… |
| 7 | Cataract | H25.0-H25.9, H26.1-H26.9 | F92 | F46…, |
| 8 | Chronic Back pain | M47-M48, M51-M54, M99, G54.4 | L02, L03, L86 | N12…, N14…, |
| 9 | Chronic kidney disease (any cause) | N02.0-N8.8, N11.0-N11.9, N12.9 N15.0-N18.9, N19.9 | U99.01 | 1Z1…, K01…, K02… |
| 10 | Chronic neck pain | M54.2 | L83… | Nyu…, N11…, N12…, N14… |
| 11 | Chronic sinusitis | J32 … | R75… | H13… |
| 12 | Chronic obstructive pulmonary diseases | J41.0-J41.8, J42.9, J43.0-J43.9, J44.0-J44.9 | R91, R95 | H3… |
| 13 | Coronary Heart Disease (Including Acute Myocardial infarction, Valvular disease, Angina), | I20.0-I25.0, I34.0-I37.0 | K74-K76 | G11…, G30…, G31… - G38… |
| 14 | Dementia | F00.0-F00.9, F01.0-F03.9, G30.0-G30.9, G31.0-G31.9 | P70-P70.02 | E00…, Eu0…, F11… |
| 15 | Depression | F32.0-F33.9 | P76 | Eu…, E11… |
| 16 | Diabetes mellitus | E10.0-E14.9 | T90…, F83.01 | C10…, F32…, |
| 17 | Dyslipidaemia (Hyper) | E78.1 | T93 | C32… |
| 18 | Eating disorders (Both) | | | Eu5…, R03… |
| 19 | Eczema/ Skin disease | L20.0-L22.9, L26.9 | S74, S87, S88 | M11… |
| 20 | Epilepsy | G40.0-G41.9 | N88 | F25… |
| 21 | Fatigue | F48.0 | A04.11 | F286… |
| 22 | Fibromyalgia | M79.7 | L18.01 | N248…, N239… |
| 23 | Gall bladder stone | K80.0-K80.8 | D98-D98.03 | 781…, J65…, 4G2…, |
| 24 | GERD (gastritis, oesophageal bleeding, duodenitis, peptic ulcer) | K21 … | D840… | J12…, J13…, J15… |
| 25 | Gastrointestinal bleeding | K25.0-K28.9 | D84-D87 | J11…, |
| 26 | Gout | M10.0-M10.9 | T92 | C34…, N023… |
| 27 | Hearing impairment (all types) | H90.0-H91.9 | H83-H86 | F59…, ZE87… |
| 28 | Heart failure | 150.0–150.9 | K77-K77.02 | G58…, |
| 29 | Hepatitis | K73.0-K73.9 | D72-D72.05 | J61…, J63… |
| 30 | HIV/AIDS | B20-B24 | B90 | A788…, A789…, AyuC… |
| 31 | Hypertension | I10.9, I11.0-I13.9, I15.0-I15.9 | K86-K87, F83.02 | G20…, G24…, G25…, G26… |
| 32 | Hyperthyroidism | E05.0-E05.9 | T85 | C02… |
| 33 | Hypothyroidism | E02.9, E03.0-E03.9 | T86 | C03…, C04… |
| 34 | Inflammatory bowel disease | K50.0-K52.9 | D94-D94.02 | J4…, |
| 35 | Irritable bowel symptoms | K58.1-K58.8 | | J52… |
| 36 | Leukaemia, lymphoma | C81.0-C86.6, C91.0-C96.9 | B72-B73 | B60…, B61…, B64… |
| 37 | Liver cirrhosis | K70.0-K71.9, K74.0-K74.6 | D97… | J615… |
| 38 | Migraine | G43.0-G43.9 | N89 | F26…, |
| 39 | Multiple sclerosis | G35.9 | N86 | F20…, |
| 40 | Osteoarthritis | M16.0-M16.9, M17.0-M17.9 | L89-L91 | N05…, |
| 41 | Osteoporosis | M80.0-M82.9 | L95… | N33…, |

**Table 3** Continued

| SI no | Conditions | ICD 10 | ICPC | Read code |
|---|---|---|---|---|
| 42 | Other blood vessel disease (Raynaud's disease, Burger's disease) | I73.0-I73.9 | K92… | G73…, |
| 43 | Parkinson's disease | G20.9 | N87-N87.01 | F12…, |
| 44 | Peripheral vascular disease (Atherosclerosis) | 170.0–170.9 | K91 | G70…, G71…, G72… |
| 45 | Polymyalgia | M35.3 | L99.12 | N20.11 |
| 46 | Psoriasis | L40.0-L41.9 | S91 | M161… |
| 47 | Psoriatic arthritis | M07.0-M07.3 | L99.13 | M160… |
| 48 | Rheumatoid Arthritis | M05.0-M05.9 | L88…, K71… | N04…. |
| 49 | Renal stones | N20.0 …. | U95 | 4G4…, 7B07…, KB12… |
| 50 | Schizophrenia and/or psychosis | F20.0-F20.9, F25.0- F25.9 | P72 | E10… |
| 51 | Severe allergy | | | H17…, SN5… |
| 52 | Sjögren's syndrome | M35.0 | NA | N002… |
| 53 | Systemic lupus erythematosus | M32.0, M32.1, M32.8, M32.9 | NA | N000… |
| 54 | Sleep disorder (Insomnia) | F51.0 | P06 | Fy0…, 1B1B… |
| 55 | Solid malignancy | C00.0-C80.9, D00.0-D09.9, C97.9 | A29…, A79, B74 – Y78 | B0… - B67…, Byu… |
| 56 | Stroke | G45.0-G46.8, I60.0-I63.9, I65.0–166.9, I69.0-I69.4 | K89-K90.02 | G60… - G68…, F22… |
| 57 | Substance abuse/drug addiction | F10.0-F19.9 | P18, P19… | E24…, Eu1… |
| 58 | Thrombotic diseases | I74.0-I74.9 | K93, K94…, W99.03 | G80…, G81…, G74… |
| 59 | Tuberculosis | A15.0-A16.9, B90.9 | A70, R70 | A1…, A11… |
| 60 | Vertigo | H81.4 … | N17-N17.02, H82-H82.03 | R004…, F561… |
| 61 | Vision problem (Glaucoma and other) | H27.0-H27.9, H40.0-H40.9, H42.0-H42.8 | F93…, F94 | F45…, F49…, |

All the codes are the primary code initials used in the database.
GERD, Gastroesophageal reflux disease; ICD-10, International Classification of Diseases, 10th edition; ICPC, International classification in primary care.

### Data harmonisation

First, we carried out a code mapping exercise for identification of people with OA and other comorbidities. A list of chronic conditions was built with different codes used by different databases for the same condition to identify eligible people with the conditions, such as Read code in Clinical Practice Research Datalink (UK), ICPC2 (Rotterdam) and ICD-10 for Lund and Spanish as used in respective database. The code lists were compared and edited to maintain the uniformity. The list was screened and verified by two researchers and two GPs. We also used uniform definition for inclusion of condition, for example, at least one recording of the chronic conditions. Because all the centres did not have all the listed comorbidities, a minimum number of chronic conditions and covariates common in all the database were identified to be studied. Similarly, we decided to have a minimum follow-up study of 5 years and centres with more registration period can use the entire length of data available.

### Study design and data analysis

Summary of the study design and analysis is provided in figure 1. All the centres plan to use same statistical analysis plan to investigate each objective.

### ETHICS AND DISSEMINATION

The study has obtained the following ethics approvals: UK- Independent Scientific Advisory Council 19/30R, The Netherlands—The Integrated Primary Care Information registration no. 11/2019, Spain—the Information System for the Development of Research in Primary Care, 4R19/011, Sweden—Ethical Review Authority, Skåne Healthcare Register, 'Dnr 2011-432, Dnr 2014-276 and Dnr 2018-233. The registry databases are made available anonymised for the research purposes. Each centre will follow the data privacy policy of respective countries.

We plan to publish all the results as manuscripts in peer reviewed journals and present the findings in relevant conferences. The results would be circulated as 'layperson' language and would be available at least four different international languages such as English, Dutch, Swedish and Spanish. These will be shared on appropriate patient–public forum and involved institution's websites.

### Objective 1: prevalence, incidence and time sequence of comorbidities in OA

A combined retrospective and prospective study of OA cases and sex, age (±2 years), first year of registration, and practice matched controls (1:1–4) without OA[28] will be

| | Objective 1 | Objective 2 | Objective 3 | Objective 4 |
|---|---|---|---|---|
| Study Design | Case-control and Cohort | Clustering and longitudinal | Cohort | Genomic association |
| Exposure | OA | OA | Analgesics (NSAIDS, Opioids, paracetamol) | OA |
| Outcome | Comorbidities | Clusters of comorbidities | Comorbidities | Comorbidities |
| Statistical methods | Conditional logistic regression, Cox regression | Latent class analysis, Latent class growth analysis Joint latent class analysis | Cox regression Time varying analysis Flexible parametric method | Mendelian randomisation |
| Reported Outcome | Odds Ratio and Hazard Ratio | Clusters and groups | Hazard Ratio | Coefficients |
| Participating centres | ALL | ALL | ALL | UK and Netherlands |

| Objective 5 | Meta Analysis |
|---|---|

**Figure 1** Overview of the study design and statistical analysis plan for the comorbidities in osteoarthritis (CoMO) study. OA, osteoarthritis.

used to determine the prevalence, incidence and time sequence of comorbidities in OA. Incident OA cases will be identified, and the first diagnosis date will be used as the starting point (index date). For controls the same index date as their matched case will be used. They will be both retrospectively reviewed for prior diagnoses of comorbidities and prospectively followed up for posterior new comorbidities. In the retrospective analysis, the prevalence and 95% CI of each specific comorbidity will be calculated in OA cases and matched controls using the number of people diagnosed with the comorbidity divided by the total number of OA cases or controls at the index date. The prevalence of each comorbidity in OA cases and matched controls will be calculated for given time intervals prior to the index date of 0–1, 0–5 and 0–10 years separately to assess observational bias.[28] Discrete time intervals of 1–5, and 5–10 years before will also be used to estimate the prevalence to minimise consultation bias/misclassification bias of OA (if possible). Logistic regression will be used to calculate the OR for each comorbidity unadjusted and adjusted for BMI, smoking and alcohol consumption.

For the prospective analysis participants with incident OA but without the specific comorbidity of interest at the index date (ie, people at risk) and matched controls without OA will be followed up until the date of the first diagnosis of the comorbidity, deregistration, or death whichever comes first. The cumulative incidence will be calculated for each comorbidity in OA cases and matched controls at 1, 3, 5, 10, 15, 20 years (based on the data available) after the index date to examine the dynamic change of developing comorbidities during follow-up. Kaplan-Meier survival curves will be used to display the cumulative probability in OA and non-OA groups. Proportional hazard assumption will be tested using Schoenfeld residual plots. The Cox regression model will be used to calculate HR for each comorbidity unadjusted and adjusted for age, sex, practice, BMI, smoking

and alcohol consumption. This hybrid design has been previously used by us to examine the temporality of associations between other rheumatic MSK diseases (eg, gout and lupus) and comorbidities.[28 29]

### Objective 2: clusters and impact of comorbidities in people with OA

For each dataset, an 80%:20% split into the training and testing data will be introduced. The following analysis in objective 2 will be first employed into the training dataset and then tested its generalisability in the testing dataset. At baseline, clusters of people based on 61 comorbidities will be identified using Latent class (ie, Gaussian mixture models algorithms of cluster) analysis.[30] For each model, we will examine the association between clusters and covariates using multinomial logistic regressions. The distinctness of clusters will be examined by comparing covariates among clusters. The optimal model is the one where most clusters found in the training data are also identified independently in the testing data and clusters have most distinct patients' characteristics. We will then use both latent trajectory analysis, such as joint latent class models,[31] and unsupervised machine learning approach, such as deep autoencoder or recurrent neural networks,[32] to identify distinct clusters of new comorbidity numbers development over time and their association with mortality with adjustment for baseline covariates.

### Objective 3: association between analgesics and incident comorbidities

A cohort study will be undertaken for this objective to evaluate the contribution of common analgesics for OA to the development of comorbidity such as NSAIDs, opioids and paracetamol. We are interested in the interaction between OA and use of drugs on the incidence of comorbidities, that is, to evaluate if the drug use in persons with OA poses increased or decreased risk of comorbidities compared to persons without OA and/or analgesics. Individual

**Table 4** Group of conditions/outcome

| Group | Conditions |
| --- | --- |
| Cardiovascular | Cardiac arrhythmias, coronary heart disease (including AMI, valvular disease, angina), heart failure, hypertension, peripheral vascular disease (claudication, Raynaud syndrome, Buerger's disease), other blood vessel disease (atherosclerosis and aneurysm), thrombotic diseases |
| Gastrointestinal (GI) | GERD (oesophageal diseases, gastritis, duodenitis), GI bleeding, inflammatory bowel disease, irritable bowel syndrome |
| Musculoskeletal | Ankylosing spondylitis, chronic back pain, chronic neck pain, fibromyalgia, polymyalgia, gout, osteoporosis, psoriatic arthritis, rheumatoid arthritis, Sjögren's syndrome, systemic lupus erythematosus |
| Endocrine | Diabetes mellitus, dyslipidaemia (hyper), hyperthyroidism, hypothyroidism |
| Neurological | Dementia, epilepsy, fatigue, migraine, multiple sclerosis, Parkinson disease, stroke |
| Psychological | Anxiety, depression, eating disorders (anorexia/Bulimia nervosa), Schizophrenia, sleep disorder (insomnia), |
| Kidney disease | Chronic kidney disease (any cause), renal stones |
| Liver diseases | Gall bladder stone, hepatitis, liver cirrhosis |
| Respiratory | Asthma, chronic obstructive pulmonary disease |
| Cancer | Leukaemia, lymphoma, solid malignancy (any type) |
| Others | Anaemia (all types), benign prostate hypertrophy, cataract, chronic sinusitis, eczema/skin disease, hearing impairment (all types), psoriasis, severe allergy (anaphylactic shock), angioneurotic oedema |

AMI, Acute Myocardial Infarction; GERD, Gastroesophageal Reflux Disease.

comorbidity, as well as clusters of comorbidities identified from objective 2 will be examined as outcomes. The 61 comorbidities in our study will be further categorised into eight groups, specifically: MSK, respiratory, neurodegenerative, psychological/psychiatric, cancer, cardiovascular, metabolic, renal problem, liver diseases, GI and others (table 4). The prospective cohort established from objective 1 will form the source population for this objective. Individuals with incident OA will be identified from the database and the first diagnosis date will be used as index date for follow-up. Individuals without OA during the study period will be selected and matched with cases by age, sex and practice. The same index date will be given from their matched OA cases. Individuals with analgesics prescriptions prior to the index date will be excluded (or recorded as a confounding factor to be adjusted as appropriate). Only analgesic prescriptions after the index date will be considered for this analysis. Prescriptions will be quantified as number of prescriptions within year 1 (initial use, primary analysis),[33] 2, 3, 4, 5, etc. It will also be dichotomised as episodic (eg, at least one gap of ≥90 days between prescriptions) and continuous (no gap of more less than 90 days) users as appropriate.[34 35] Analgesic use will be included in the model as a risk factor together with OA diagnosis (yes/no, primary exposure) to examine the independent risk of each variable (OA and analgesics), as well as the interaction between the two to the development of comorbidity. Dose response relationship will be examined using number of prescriptions during the exposure window examined. The effect of stopping analgesics will also be examined by looking into the patterns of analgesic prescriptions, for example, stopping analgesics after initial use in year 1 versus continuous use of analgesics afterwards. For the primary analysis (initial prescriptions within year 1), a landmark analysis will be used to minimise the immortal time bias where the follow-up will start after 12 months from the index date.[36] Participants at risk (ie, without a specific comorbidity of interest) at the landmark date will be followed up until the first diagnosis of the comorbidity, deregistration or death whichever comes first. For secondary analyses, time varying covariate analysis will be used to examine the long-term, episodic/continuous use of analgesics after the index date and interaction between OA and analgesics in the development of the comorbidity. The propensity score matching or the inverse probability weighing methods will be used to adjust for confounding by indication during the follow-up as appropriate. Depending on the country-specific drug use patterns, we may modify this definition to allow for short brakes in between the episodes. Cox-regression model will be used to calculate the HR and 95% CI. We will use flexible parametric models using restricted cubic splines (developed by Lambert, 'stpm2' in Stata) to estimate the HRs and differences in time to diagnosis of comorbidities (outcome) with drugs as time-varying to account for non-proportional hazards.[37]

### Objective 4: potential causal pathways between OA and the comorbidities

We will perform an MR Phenome-Wide Association Study (MR-PheWAS)[38] to examine the causal relationship between OA, its phenotypes, biomarkers or risk factors and comorbidities using the UK Biobank and the Rotterdam Study database.

We will use the Rotterdam Study and the UK Biobank jointly for this objective. This is because that the Rotterdam Study is an OA cohort with deep phenotypes and biomarkers of OA, whereas the UK Biobank is a primary cohort for cancer and multiple disease

outcomes, and both have detailed genetic variants. We will use two sample MR approach, that is, to establish an association between OA and genetic variants in the Rotterdam Cohort to identify genetic IV, for example, a set of single nucleotide polymorphisms associated with OA (or a deep phenotype, biomarker or risk factor of OA). We will then undertake the MR-PheWAS analysis to examine the causal effects of the OA IV on comorbidities in the UK Biobank. The MR method has been widely used in real world data to examine the causal relationship between IV and specific disease, under two assumptions: (1) genetic variants are randomly assigned in the population and (2) genetic variants can only be the cause not consequence of disease.[39] The PheWAS is a series of case control studies to estimate the associations between the IV and multiple disease outcomes.[38 40] The combination of the two permits investigation of the causal effects of OA on multiple disease outcomes.

The MR-PheWAS analysis includes three steps. First, we will identify the genetic variants that are associated with OA-IV. Second, we will undertake the PheWAS analysis—a series of case control analyses to estimate the associations between the IV and other disease outcomes,[38 40] with an adjustment for multiple testing using the false discovery rate methods.[41] Third, we will implement conventional MR analysis to investigate the causal effects of the OA IV on comorbidities.[39] An inverse variance weighted method will be used to pool the associations (ORs) as appropriate.[42] The MR-Egger regression analysis will be used to count for the pleiotropic effect—the effects of one genetic variant on multiple outcomes.[43] The heterogeneity in dependent instruments test will be used to exclude the cross-phenotype associations caused by genetic linkage.[44]

With the MR-PheWAS study, the OR can be interpreted as causal association. We are primarily interested in the causality from OA to comorbidities. We are also interested in inflammatory (eg, C reactive protein), metabolic (eg, gut microbiome) and biomechanics (eg, BMI) biomarkers and deep phenotypes of OA such as knee, hip and hand OA with and without symptoms. This will be undertaken if it is feasible within 3 years of this funded project, otherwise will be considered as our future research agenda.

### Objective 5: variation of OA comorbidity patterns across countries

We will use meta-analyses to examine the variation between countries and to pool the data as appropriate. Estimates from first three objectives such as prevalence, incidence, OR, HR and 95% CI for each specific comorbidity across different populations will be distributed in a forest plot. Heterogeneity will be examined using the $I^2$ statistic and the Q test.[45] Results will be pooled if they are homogenous based on the $I^2$ value using the fixed effects model, otherwise the reasons for the heterogeneity will be investigated. Random effects models will be used to pool the results if the reasons for the heterogeneity cannot be identified and if the overall pooling is appropriate.

Individual patient data meta-analysis may be used to help identify the reasons for heterogeneity.[46] Common clusters and trajectories as well as burdens of comorbidities will also be compared between populations.

### Feasibility and sample size

To detect minimum incidence of 1% comorbidity (required for cluster analysis) with a minimum clinical important difference of HR1.2, and 90% power of the study, the estimated sample size was 197 561 for 1581 events. It was calculated using STATA v.15, with a correlation=0.2, SD of 0.5, proportion of withdrawal=0.20, alpha=0.05. The initial check with the registry database revealed to have minimum required sample size for the study.

### DISCUSSION

This study will be the largest epidemiological study on comorbidities of OA in primary care. One of the key advantages of this multinational study is the use of the same protocol to measure the burden of comorbidities in primary care settings in four European countries to ensure reproducibility and comparison. There is scant evidence on the comorbidities in people with OA, and this approach should help to identify the leading and most important associations before and after presenting clinical OA (the index date). Further advantages of this study are the large and representative populations studied and the same/similar extensive list of chronic conditions for identifying comorbidity clusters. Often comorbidities accumulate with age over time and the large primary care databases in this study have the advantage of having long follow-up time which will enable us to detect the incidence of comorbidities. Also, longer follow-up would help to identify the picture of the trajectory of the diseases.[47] Both the incidence and the trajectories of comorbidity clusters are highlighted as key elements needed in current research in multimorbidity, so findings from this study should help to fill the knowledge gaps on multimorbidity in OA.

The relationship between chronic conditions and polypharmacy is a complex area of research. The count of the medications and more importantly the nature of prescribed drugs may be responsible for developing many new comorbidities in people with OA. We aim to explore the associations of the most commonly prescribed drugs in OA, such as NSAIDs, with the incidence of a wide range of comorbidities, which will be the first time that conditions other than established comorbidities such as psychological conditions and endocrine diseases will be examined. Finally, the causality study will further explore the associations at genetic levels and phenotypes, which will be novel in OA research. Using a two sample MR approach—one for OA deep phenotypes and the other for other chronic conditions maximises the potentials of sample size, disease phenotypes and comorbidity

spectrum to better explore the causal pathways between OA and comorbidity.

There are some limitations to this study. First, there are inherent issues in the nature of electronic health records with respect to possible misdiagnosis, ascertainment biases, under-recording, and changes in databases due to change in coding structures. Also, the analysis will be restricted to fewer covariates in some databases due to missing information on lifestyle factors such as physical activities and diet. Even though the databases have different durations of data available, if possible we will use a common follow-up time for objective five. Another important limitation is that we do not have information on quality of life and other outcomes to measure functional limitations recorded in the database.

Chronic conditions, especially comorbidities recorded in general practices, depend on multiple factors such as population structure, healthcare facilities, health policies and the nature of the national databases. A major strength of this study is that it will include medical records on approximately 27 million people in four European countries. Also, the study will cover the sequence of research questions in comorbidity or multimorbidity starting from the burden through to the causality and variation. Such a research can be used for other similar multimorbidity studies. The expected results should inform health professionals in primary care settings with respect to management of people with OA and associated comorbidities.

## Status of the study

All the centres have obtained the necessary approvals for using the database in 2020. A consensus has been made on the code mapping exercise. The statistical analysis will be explained in detail in each of the publications. The team is expected to produce results by mid-2021.

**Author affiliations**
[1]Academic Rheumatology, University of Nottingham School of Medicine, Nottingham, UK
[2]Nuffield Department of Primary Care Health Sciences, University of Oxford, Oxford, UK
[3]Department of General Practice, Erasmus MC University Medical Center Rotterdam, The Netherlands, Rotterdam, The Netherlands
[4]Department of Clinical Sciences, Clinical Epidemiology Unit, Orthopaedics, Lund University, Lund, Sweden
[5]Center for Statistics in Medicine, Nuffield Department of Orthopaedics Rheumatology and Musculoskeletal Sciences, University of Oxford Nuffield, Oxford, UK
[6]School of Medicine, Keele University, Keele, UK
[7]Division of Rheumatology, Allergy, and Immunology, Chang Gung Memorial Hospital, Taoyuan, Taiwan
[8]Division of Primary Care, University of Nottingham, Nottingham, UK
[9]Pain Centre Versus Arthritis, University of Nottingham, Nottingham, UK
[10]Musculoskeletal Research Unit, Bristol Medical School, Translational Health Sciences, University of Bristol, Bristol, UK
[11]Department of General Practice, Department of Orthopaedic Surgery & Sports Medicine, Erasmus MC University Medical Centre Rotterdam, Rotterdam, The Netherlands

**Acknowledgements** We thank the Patient Research Participants (PRP) members Jenny Cockshull, Stevie Vanhegan and Irene Pitsillidou for their involvement throughout the project. We would like to thank the FOREUM for financially supporting the research. The authors would like to acknowledge Keele University's Prognosis and Consultation Epidemiology Research Group who have given us permission to utiliseuse the Code Lists (2014).

**Contributors** WZ, MD, CC, SMAB-Z, ME and DP-A conceived and designed the study. SS, AK, AD, AT, DR and VS developed the methods and will perform the analysis and interpretation of the results. CC, WZ, JR, AS, C-FK, VS and AT will supervise the statistical analysis. CM and MD will guide with clinical interpretations of the results. SS drafted this manuscript and all authors contributed to the critical revision of the manuscript for important intellectual content. The corresponding author attests that all listed authors meet authorship criteria and that no others meeting the criteria have been omitted.

**Funding** This work was supported by Foundation for Research in Rheumatology (FOREUM) grant (2019-2022), The Swedish Research Council (2020-01103), Governmental funding of clinical research within the national health services (ALF), and The Swedish Rheumatism Association. CM is funded by the National Institute for Health Research (NIHR) Applied Research Collaboration West Midlands, the National Institute for Health Research (NIHR) School for Primary Care Research and a National Institute for Health Research (NIHR) Research Professorship in General Practice (NIHR-RP-2014-04-026).

**Disclaimer** The sponsors did not participate in the design and conduct of the study; collection, management, analysis, and interpretation of the data; or preparation, review, or approval of the manuscript and the decision to submit the manuscript for publication.

**Competing interests** WZ declares serving as an advisory board for Ely Lilly (Ixekizumad, 2020) and Regeneron (Fasinomab, 2020). ME declares serving as an advisory Panel Board Member for Pfizer (Nov 2019, Tanezumab). CM provided advice to BMS on recruiting to a non-pharmacological atrial fibrillation trial.

**Patient and public involvement** Patients and/or the public were involved in the design, or conduct, or reporting, or dissemination plans of this research. Refer to the Methods section for further details.

**Patient consent for publication** Not applicable.

**Provenance and peer review** Not commissioned; externally peer reviewed.

**ORCID iDs**
Subhashisa Swain http://orcid.org/0000-0001-9207-1065
Andrea Dell'Isola http://orcid.org/0000-0002-0319-458X
Aleksandra Turkiewicz http://orcid.org/0000-0003-1460-2275
Danielle Robinson http://orcid.org/0000-0002-0940-9211
Chang-Fu Kuo http://orcid.org/0000-0002-9770-5730
Daniel Prieto-Alhambra http://orcid.org/0000-0002-3950-6346

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
