## [Reviewer comments · BMJ Open]

ARTICLE DETAILS

TITLE (PROVISIONAL)	Comorbidities in Osteoarthritis (ComOA): a combined cross-sectional, case-control and cohort study using large electronic health records in four European countries
AUTHORS	Swain, Subhashisa; Kamps, Anne; Runhaar, Jos; Dell'Isola, Andrea; Turkiewicz, Aleksandra; Robinson, Danielle; Strauss, V; Mallen, Christian; Kuo, Chang-Fu; Coupland, Carol; Doherty, Michael; Sarmanova, Aliya; Prieto-Alhambra, Daniel; Englund, Martin; Bierma-Zeinstra, Sita; Zhang, Weiya

VERSION 1 – REVIEW

REVIEWER	Ock, Minsu University of Ulsan College of Medicine
REVIEW RETURNED	18-May-2021

GENERAL COMMENTS	I applaud the authors' effort in constructing the protocol of a multinational study in four European countries (UK, Netherlands, Sweden, and Spain) exploring comorbidities in people with OA. It is clearly an important area of research, and the manuscript is structured and easy to read. I look forward to various studies regarding OA patients using the constructed database. However, one question is whether the health-related quality of life information for OA patients was collected. For OA patients, health-related quality of life is expected to be available as an important outcome indicator. Thank you for the opportunity to read your manuscript.
---

REVIEWER	Parkinson, Lynne The University of Newcastle, School of Medicine and Public Health
REVIEW RETURNED	01-Dec-2021

GENERAL COMMENTS	General This is an excellent idea and presented as deceptively simple. Harmonising across national databases, let alone across countries is an enormous task. The methods proposed are generally well thought out and sophisticated, but some more detail would be appreciated in places, as noted. I am concerned about the feasibility of subgroup sample sizes, given the incidence and number of covariates. The results will be of great value. Specific Introduction Quite well argued, however, the argument is a little hurried. An heuristic for writing an introduction is one idea per sentence and one message per paragraph. I encourage the authors to rewrite using this as a guide. Language needs proofing. Methods
--

	I appreciate the detail provided in Tables 3 and 4, although I suspect this is the start of a very long harmonising journey. I would like to see some more detail about how harmonisation across databases will be achieved-what are your guidelines? Do you have a model to use? I am a bit concerned about “Missing data on covariates will be substituted using multiple imputation methods, provided that the data is missing at random, if applicable.” (page 7, line 2) Perhaps you need more detail here. I would be happier with an assumption that if not recorded previously, it is considered not present, and if it has been recorded previously, then it is assumed to be present. For objective 2, I would like to see estimated sample sizes, to understand feasibility. Machine learning algorithms generally require big sample sizes. There is a risk that some sample sizes will be very small when diluted by low incidence comorbidities. Limitations I would like to see the issue of database harmonisation dealt with in more detail.
--	---

VERSION 1 – AUTHOR RESPONSE

Reviewer: 1

Dr. Minsu Ock, University of Ulsan College of Medicine

Comments to the Author:

Q1. I applaud the authors' effort in constructing the protocol of a multinational study in four European countries (UK, Netherlands, Sweden, and Spain) exploring comorbidities in people with OA. It is clearly an important area of research, and the manuscript is structured and easy to read. I look forward to various studies regarding OA patients using the constructed database. However, one question is whether the health-related quality of life information for OA patients was collected. For OA patients, health-related quality of life is expected to be available as an important outcome indicator. Thank you for the opportunity to read your manuscript.

Author's Response: We agree that this is a very interesting aspect of OA to be studied. However, we will use primary care databases from four different countries, which capture each encounter of people in primary care. Unfortunately, no such databases do not have any recording of quality of life.

Author action: We have added this as a limitation to our study.

“In the databases there is no information on quality of life and other outcomes to measure functional limitations.” Page 12, lines 363-364.

Reviewer: 2

Prof. Lynne Parkinson, The University of Newcastle

Comments to the Author:

Q1. General

This is an excellent idea and presented as deceptively simple. Harmonising across national databases, let alone across countries is an enormous task. The methods proposed are generally well thought out and sophisticated, but some more detail would be appreciated in places, as noted. I am concerned about the feasibility of subgroup sample sizes, given the incidence and number of covariates. The results will be of great value.

Author Response: We have explained the details in the following specific sections.

Specific

Q2. Introduction

Quite well argued, however, the argument is a little hurried. An heuristic for writing an introduction is one idea per sentence and one message per paragraph. I encourage the authors to rewrite using this as a guide. Language needs proofing.

Author Response: Thank you for the suggestions. We have revised the introduction as suggested .

Q3. Methods

I appreciate the detail provided in Tables 3 and 4, although I suspect this is the start of a very long harmonising journey. I would like to see some more detail about how harmonisation across databases will be achieved-what are your guidelines? Do you have a model to use?

Author Response: We agree with the reviewer's comment. We have now explained the following steps used for data harmonisation.

Author Action: Data Harmonisation:

Firstly, we carried out a code mapping exercise for the identification of people with osteoarthritis (OA) and other comorbidities. We developed a list of chronic conditions and each centre shared the list of codes to be used for the conditions, such as Read code in CPRD (UK), ICPC-1 (Netherlands) and ICD-10 for Sweden and ICD-10CM for Spain, as per their database. The code lists were compared and edited to maintain uniformity. The list was screened and verified by two researchers and two GPs. We also used uniform definitions for inclusion of conditions e.g. at least one recording of the chronic conditions. Because not all centres had all the listed comorbidities, a minimum number of chronic conditions and covariates common in all the database were identified to be studied. Similarly, we decided to have a minimum database follow-up of 5 years and that centres with a longer registration period could use the entire length of data available." Page 7, Lines 183-194

Q4. I am a bit concerned about "Missing data on covariates will be substituted using multiple imputation methods, provided that the data is missing at random, if applicable." (page 7, line 2) Perhaps you need more detail here. I would be happier with an assumption that if not recorded previously, it is considered not present, and if it has been recorded previously, then it is assumed to be present.

Author Response: We agree with the simpler explanation of the missing data. The purposes of missing at random is for objective 2, which requires an assumption for performing the trajectory analysis. However, we have changed our statement as suggested.

Author Action: "If the data are not recorded, they will be considered as not present." Page 6, Lines 176-177

Q5. For objective 2, I would like to see estimated sample sizes, to understand feasibility. Machine learning algorithms generally require big sample sizes. There is a risk that some sample sizes will be very small when diluted by low incidence comorbidities.

Author Response: We have added a section on sample size focusing on the cohort study, primary aim of the study. Low-prevalent comorbidities (as long as prevalence is above 1-2%, which it is for most of our conditions) do not need to be a problem when deriving clusters. Also, there are two criteria for prevalence and correlation that have to be met for cluster analysis in objective 2.

Author Action: "To detect minimum incidence of 1% comorbidity (required for cluster analysis) with a minimum clinically important difference of hazard ratio (HR)1.2, and 90% power of the study, the estimated sample size was 197561 for 1581 events. It was calculated using STATA, with a correlation= 0.2, standard deviation of 0.5, proportion of withdrawal= 0.20, alpha=0.05. The initial check with the registry database revealed to have minimum required sample size for the study." Page 11-12, Lines 346-347.

"Before using LCA, the prevalence of the comorbidities will be measured, and those with less than 1% will be excluded. Then, correlations among variables will be checked and, for any pair $> \pm 0.6$, clinical discussion will be taken to decide which (or both) are going to be used in the next steps." Page 8, Lines 239-242.

Q6. Limitations

I would like to see the issue of database harmonisation dealt with in more detail.

Author Response: We have explained details of data harmonisation in response to Q3.

Author Action: In addition, "All the centres plan to use the same statistical analysis plan to investigate each objective." Page 7, Lines 196-197

VERSION 2 – REVIEW

REVIEWER	Parkinson, Lynne The University of Newcastle, School of Medicine and Public Health
REVIEW RETURNED	11-Jan-2022
GENERAL COMMENTS	Thank you for your response. I am satisfied that all my concerns have been answered.

VERSION 2 – AUTHOR RESPONSE

Reviewer: 2

Prof. Lynne Parkinson, The University of Newcastle Comments to the Author:

Thank you for your response. I am satisfied that all my concerns have been answered.

Author's response- Thank you.